# Isolation, Identification, and Pathogenicity of an Avian Reovirus Epidemic Strain in Xinjiang, China

**DOI:** 10.3390/v17040499

**Published:** 2025-03-30

**Authors:** Xin Ma, Weiqi Li, Zhaoquan Liu, Zhipeng Zuo, Xinyu Dang, Hengyun Gao, Qingling Meng, Lin Yang, Yongjie Wang, Shilei Zhang

**Affiliations:** 1College of Animal Science and Technology, Shihezi University, Shihezi 832003, China; 2Department of Animal Sciences, North Carolina Agricultural and Technical State University, Greensboro, NC 27411, USA

**Keywords:** avian reovirus, broilers, isolation, identification, pathogenicity

## Abstract

To investigate the prevalence and pathogenic characteristics of avian reovirus (ARV) in the Xinjiang region of China, this study collected suspected joint tissue samples from broiler farms across different areas of the Taikun Group. The samples were subjected to virus isolation, RT-PCR analysis, sequence analysis, in vitro replication assays, and pathogenicity assessments in specific pathogen-free (SPF) chicken embryos and chickens. The results revealed the isolation of a chicken-derived ARV epidemic strain, designated as ARV xj-1.1. The virus-induced cytopathic effects (CPEs) in LMH cells and the time required to observe CPEs significantly decreased with serial blind passages. Pathogenicity testing demonstrated that ARV xj-1.1 is highly virulent to SPF chicken embryos and chickens. Inoculation of SPF chicken embryos resulted in a 100% mortality rate, while inoculation of 1-day-old SPF chicks caused significant swelling of the footpads. In conclusion, this study successfully isolated an epidemic strain of avian reovirus, providing a valuable theoretical basis for understanding the genetic evolution and epidemiology of ARV variants in the Xinjiang region.

## 1. Introduction

In recent years, cases of viral arthritis and tenosynovitis caused by avian reovirus (ARV) infection have increased sharply worldwide, leading to significant economic losses in the poultry industry [1,2]. ARV was first isolated from diseased chickens by Fahey and Crawley in 1954 and primarily infects poultry, particularly commercial broilers [3]. ARV infection can result in a morbidity rate as high as 100%, though its associated mortality rate remains relatively low [4]. Clinical manifestations of ARV infection include viral arthritis, respiratory diseases, enteritis, and growth retardation, categorizing it as an immunosuppressive disease [4,5,6,7]. The susceptibility of chickens to ARV is age-dependent, with 1- to 2-week-old chicks being the most vulnerable, displaying severe clinical symptoms [8]. In contrast, adult chickens often experience a prolonged incubation period with milder clinical signs [9].

ARV belongs to the genus Orthoreovirus within the family Reoviridae [10]. The virus is a non-enveloped, double-stranded RNA virus with an icosahedral capsid structure, measuring approximately 70–80 nm in diameter [11]. The ARV genome comprises ten segments, categorized based on their electrophoretic mobility into L1, L2, L3, M1, M2, M3, S1, S2, S3, and S4 [12]. These segments encode four non-structural proteins and eight structural proteins [5]. Due to its segmented genome, ARV is prone to genetic mutations and reassortment, particularly in the S1 segment, which encodes the highly variable σC protein. The S1 sequence is widely utilized for ARV genotyping, classifying the virus into six distinct genotypes [13]. Notably, commercial vaccine strains currently available belong to genotype I. However, the emergence of novel ARV field strains has rendered existing vaccines insufficient for providing full protection, posing a significant challenge to poultry health management. To gain a deeper understanding of the prevalence and genetic diversity of ARV in Xinjiang, China, this study successfully isolated an ARV strain from joint tissue samples collected from broiler farms in different regions of the Taikun Group. The findings aim to provide molecular epidemiological insights that can support the development of improved prevention and control strategies against ARV infections in this region.

## 2. Materials and Methods

### 2.1. Reagents

A DMEM/F12 culture medium (Gibco, 8122583, New York, NY, USA), fetal bovine serum (ExCeLL Bio, FSP500, Suzhou, China), penicillin–streptomycin solution (Solarbio, P1400, Beijing, China), phosphate-buffered saline (PBS) (Solarbio, P1020, Beijing, China), RNA extraction kits (TransGen Biotech, ER501-01, Beijing, China), reverse transcription kits (Takara, RR047A, Dalian, China), 1 × Taq PCR MasterMix Purple (Biomed, MT261-03, Beijing, China), and a DL2,000 DNA Marker (Takara, 3427A, Dalian, China) were used in this study.

### 2.2. Cells and Experimental Animals

LMH chicken hepatocellular carcinoma cells (Beyotime, C7780, Shanghai, China), specific pathogen-free (SPF) chicken embryos, and SPF chickens (Haotai, Linyi, Shandong, China) were included.

### 2.3. LMH Cell Recovery

Frozen LMH cells stored in liquid nitrogen were rapidly retrieved and immediately thawed in a 37 °C water bath with gentle shaking. The cells were centrifuged at 1000 rpm for 5 min, and the cryovial was transferred under aseptic conditions to a biosafety cabinet. The supernatant was discarded, and the cell pellet was resuspended in 1 mL of DMEM/F12 medium supplemented with 10% fetal bovine serum. The suspension was transferred into a T-25 cm^2^ cell culture flask containing 7 mL of complete medium and evenly distributed using a figure-eight motion. The cells were incubated at 37 °C with 5% CO_2_ and observed daily for growth and morphology.

### 2.4. Virus Isolation and Propagation

Joint fluid and cartilage samples were collected from suspected Avian Reovirus (ARV)-infected yellow-feather broilers from various farms in the Taikun Group, Xinjiang. The samples were ground in liquid nitrogen, transferred into 1.5 mL centrifuge tubes, and homogenized with 1 mL PBS. The suspension was centrifuged at 10,000 rpm, and the supernatant was filtered through a 0.22 µm membrane. The filtered virus solution (8.62 × 10^7^ copies/μL, 1 mL) was inoculated into a pre-cultured monolayer of LMH cells in a T-25 cm^2^ culture flask. After incubation for 1 h at 37 °C in a 5% CO_2_ incubator (311, Thermo Fisher Scientific, Waltham, MA, USA), the virus-containing medium was discarded and replaced with DMEM/F12 supplemented with 2% fetal bovine serum to maintain cell viability. Cells were observed for cytopathic effects (CPEs) over five consecutive days. When 80% of the cells exhibited CPEs, they were collected, subjected to repeated freeze–thaw cycles, and serially passaged five times.

### 2.5. Virus Identification by RT-PCR

Viral RNA was extracted from the supernatant of infected cell cultures using an RNA purification kit (TransGen Biotech Co., Ltd., Beijing, China). Reverse transcription was performed using the PrimeScript RT kit (TransGen Biotech Co., Ltd., Beijing, China). Based on the S3 gene sequence of the S1133 ARV standard strain, PCR primers were designed using NCBI Primer-BLAST (https://www.ncbi.nlm.nih.gov/tools/primer-blast/index.cgi?GROUP_TARGET=on, assessed on 15 December 2024). The primer sequences were as follows: ARV-F: TGCAAGCCGCAATGGAGGT and ARV-R: GACCCGGAGGTACTTACCAAC, with a target fragment size of 1128 bp. PCR amplification (DYY-2C, Thermo Fisher Scientific, Waltham, MA, USA) was performed in a 25 µL reaction system containing 0.5 µL of each primer (10 µmol/L), 2 µL of cDNA template, and 1× PCR Mix. The reaction conditions included an initial denaturation at 94 °C for 5 min, followed by 34 cycles of denaturation at 94 °C for 20 s, annealing at 53 °C for 30 s, and extension at 72 °C for 90 s, with a final extension at 72 °C for 10 min. PCR products were analyzed via electrophoresis on a 1% agarose gel (DYCP-31DN, Liuyi, Beijing, China) and photographed (FL1500, Thermo Fisher Scientific, Waltham, MA, USA).

### 2.6. Quantitative PCR (qPCR) Identification of the Virus

PCR products containing the target bands were sequenced by Beijing Ruibo Xingke Biotechnology Co., Ltd. (Beijing, China). The sequencing results were analyzed using BLAST (https://www.ncbi.nlm.nih.gov/) (assessed on 15 December 2024). Based on the sequenced σC subunit of the S1 segment, qPCR primers were designed as follows: σC-F: 5′-TGACGTCGTATTCTGCCGAC-3′ and σC-R: 5′-CCAGCACATGGAATCAAGCG-3′, with a target fragment size of 241 bp. PCR products were validated via conventional PCR to confirm specificity. The qPCR reaction was performed in a 20 µL system containing 10 µL of 2 × Taq Pro Universal SYBR qPCR Master Mix, 0.4 µL of each primer, 2 µL of cDNA template, and 7.2 µL of ddH_2_O. The qPCR cycling conditions included an initial denaturation at 95 °C for 30 s, followed by 40 amplification cycles of 95 °C for 10 s and 62 °C for 30 s. The melting curve analysis was performed with a program consisting of 95 °C for 15 s, 60 °C for 60 s, and 95 °C for 15 s.

### 2.7. Transmission Electron Microscopy (TEM) of ARV

The Fifth Generation of the Virus (F5) viral supernatant was added to a pre-seeded monolayer of LMH cells in a T-25 cm^2^ flask. After three freeze–thaw cycles at −80 °C, the viral suspension was collected and centrifuged at 4 °C, 4500 rpm for 20 min. The supernatant was filtered through a 0.22 µm membrane and ultracentrifuged at 200,000× *g* for 4 h. The resulting pellet was resuspended in 400 µL of sterile PBS and stored in aliquots at −80 °C. For TEM observation, copper grids were immersed in the viral suspension for 1–2 min, blotted with filter paper, stained with 1% phosphotungstic acid for 1 min, and air-dried before examination under an electron microscope.

### 2.8. Viral Replication Kinetics In Vitro

The F5 viral supernatant was serially diluted 10-fold in DMEM/F12, resulting in dilutions ranging from 10^−1^ to 10^−10^. Each dilution was tested in eight replicates. LMH cells were plated in 96-well plates and infected with 100 µL of viral suspension per well, followed by incubation at 37 °C for 1 h. After removing the inoculum, cells were cultured in 200 µL of DMEM/F12 containing 2% fetal bovine serum for five days, and cytopathic effects were recorded. The median tissue culture infectious dose (TCID_50_) was calculated using the Reed–Muench method [14]. For growth kinetics, LMH cells in T-25 cm^2^ flasks were infected at 10^4.6^ TCID_50_/0.1 mL, and supernatants were collected at 12, 24, 36, 48, 60, and 72 h post-infection for TCID_50_ determination. The control group was LMH cells without infection. A one-step viral growth curve was generated.

### 2.9. Pathogenicity Assessment in SPF Chicken Embryos and Chicks

SPF chicken embryos (9–11 days old) were randomly divided into two groups (n = 10 each). The F5 viral supernatant was filtered through a 0.22 µm membrane and inoculated onto the chorioallantoic membrane at a dose of 0.2 mL (10^4.6^ TCID_50_/0.1 mL). The control group received an equal volume of sterile saline. Embryos were incubated at 37 °C and monitored daily for viability. For SPF chick footpad infections, ten 1-day-old SPF chicks were randomly divided into two groups (n = 5 per group). The experimental group received a 0.1 mL injection of viral suspension (10^4.6^ TCID_50_/0.1 mL) into the left footpad, while the control group received an equal volume of sterile saline. Chicks were observed daily for clinical signs and footpad swelling for 7 days. The clinical signs and footpad swelling on Days 1, 3, 5, and 7 were collected and reported.

## 3. Results

### 3.1. Virus Isolation and Cultivation

The virus suspension was inoculated into LMH cells, and the cytopathic effects (CPEs) were observed. Normal LMH cells exhibited a polygonal, rhomboid-like shape with uniform cytoplasm. At 24 h post-infection (hpi) with ARV, cytopathic changes began to appear, characterized by cell rounding and detachment, leading to the formation of syncytia. By 96 hpi, a large-scale detachment of cells was observed, resulting in the formation of syncytial clusters suspended in the culture medium (Figure 1).

### 3.2. RT-PCR Identification of ARV Virus

The electrophoresis results of PCR amplification using the specific primers designed in this study showed a successfully amplified specific band of approximately 1128 bp from the F5 generation culture supernatant (Figure 2). Sequence alignment of the σC gene of the isolated strain using NCBI BLAST (https://www.ncbi.nlm.nih.gov/tools/primer-blast/index.cgi?GROUP_TARGET=on, assessed on 15 December 2024) revealed homology with the σC gene sequences of the following strains: Reo/Breeder/JNSH/150826 (accession number: MK189481; 80.95%), Reo/USA/Broiler/1057NY/18 (accession number: MW854823; 79.76%), ARV 126783 clone2 (accession number: MT127456; 79.65%), Reo/0316/LN/2016 (accession number: PQ324596; 79.57%), and ARV 131495 (accession number: MT127453; 79.57%) (Figure 3). These findings confirm the successful isolation of an ARV epidemic strain, which has been designated as ARV xj-1.1.

### 3.3. Fluorescent Quantitative PCR Identification of ARV Virus

The sequence alignment of the σC gene of the ARV xj-1.1 strain with the vaccine strain ARV S1133 is shown in Figure 4. Based on the alignment results, specific RT-qPCR primers were designed (targeting the 497–737 bp region). PCR amplification was performed using cDNA from ARV xj-1.1 and ARV S1133 as templates. Electrophoresis results revealed a specific band of 241 bp for ARV xj-1.1, while no band was observed for the ARV S1133 vaccine strain (Figure 5). The fluorescent quantitative PCR (RT-qPCR) amplification curve showed a CT value of 26.51 ± 0.31, with good reproducibility and stability, and no non-specific amplification was observed (Figure 6). Under the annealing temperature of 64 °C, the RT-qPCR melting curve also exhibited high reproducibility and no signs of non-specific amplification (Figure 7).

### 3.4. Transmission Electron Microscopy (EM) Observation

The ARV epidemic strain was subjected to repeated freeze–thaw cycles, followed by centrifugation to collect the supernatant. The virus particles were then ultrafiltered, concentrated, negatively stained, and observed under a transmission electron microscope. The observed viral particles had a diameter of approximately 70 nm, displaying a spherical shape with a double-layered membrane structure (Figure 8), consistent with the characteristics of avian reovirus.

### 3.5. In Vitro Replication Kinetics of the Virus

The replication kinetics of the ARV epidemic strain were assessed by determining the TCID_50_ titers at different time points post-infection in LMH cells. The viral growth curve was plotted based on these measurements (Figure 9). The results showed that the viral titer increased gradually with incubation time, reaching its highest replication rate between 48 and 60 h post-infection, indicating the logarithmic growth phase. The peak viral titer was observed at 60 h post-infection, reaching 10^10.5^ TCID_50_/0.1 mL. After 72 h, a slight decline in viral titer was detected. These findings confirm that the ARV epidemic strain efficiently replicates in LMH cells, demonstrating high replication efficiency, with the highest viral titer recorded at 60 h post-infection.

### 3.6. Pathogenicity of the ARV Isolate in Chicken Embryos and SPF Chickens

Between 72 and 120 h post-inoculation, ARV infection induced congestion and hemorrhage in the superficial blood vessels of SPF chicken embryos, ultimately leading to embryo mortality. Due to systemic hemorrhaging, the entire embryo appeared dark red, exhibiting tissue necrosis and developmental abnormalities (Figure 10). In one-day-old SPF chickens, footpad inoculation with the virus resulted in pronounced swelling and edema in the footpads, with a morbidity rate of 100% (Figure 11). In contrast, no signs of disease or mortality were observed in the control group throughout the experimental period.

## 4. Discussion

Since the global emergence of avian reovirus (ARV), both inactivated and live attenuated vaccines have been widely used to control the disease. However, in recent years, ARV has exhibited increasing genetic diversity and enhanced pathogenicity, leading to more frequent outbreaks in poultry farms [15]. Infected birds often present with joint swelling, lameness, and growth retardation, which cause significant economic losses in the poultry industry [10,16]. Current commercial vaccine strains, including S1133, 1733, and 2408, are increasingly ineffective against emerging ARV variants, highlighting the urgent need for continued surveillance and vaccine development [3,17,18,19]. Recent studies have reported high genetic variability among ARV field strains, leading to the emergence of novel genotypes. Farkas et al. [20] analyzed ten ARV strains isolated from young chickens in Hungary (2002–2011) and found evidence of heterologous recombination in the μB gene, suggesting that recombination plays a critical role in ARV evolution alongside point mutations and reassortment. Similarly, Zanaty et al. [21] detected novel group IV and V ARV genotypes in Egypt, with low sequence homology (43–55%), indicating that these strains had recently emerged in Egyptian poultry farms. In China, Chen et al. [22] investigated ARV strains associated with arthritis/tenosynovitis syndrome in broiler flocks across Shandong Province. Their analysis identified genotypes II and V, which had not been previously reported in China, suggesting that these variants may be spreading. More recently, Yan et al. [23] used next-generation sequencing (NGS) to isolate and characterize a recombinant genotype II ARV strain from a broiler in China. Whole-genome sequencing revealed high variability in the σC gene, suggesting that segment recombination, intrasegmental recombination, and point mutations contributed to its emergence [20].

Beyond commercial poultry, ARV has also been detected in wild birds. In 2023, Zhu et al. [24] reported the first ARV infection in black swans in China, highlighting the potential role of wild birds in ARV transmission and the need for continued surveillance. With an increasing number of ARV variants emerging, current commercial vaccines are losing efficacy, providing incomplete protection against circulating strains. This underscores the importance of continued monitoring, genetic characterization, and vaccine development to combat ARV. In this study, we successfully isolated and identified an ARV strain, designated ARV xj-1.1, from a broiler farm in Xinjiang, China. LMH cells were highly susceptible to this strain, developing typical cytopathic effects (CPEs), including cell rounding, fusion, and detachment. Notably, as the virus underwent serial blind passages, the time required for CPE manifestation decreased significantly, suggesting enhanced viral adaptation and replication efficiency. Growth kinetics analysis in LMH cells demonstrated that ARV xj-1.1 replicated rapidly, reaching a peak titer of 10^10.5^ TCID_50_/0.1 mL at 60 h post-infection. This high replication efficiency further confirms the strong infectivity and adaptability of the isolated strain. To confirm the genetic identity of ARV xj-1.1, we performed RT-PCR and qPCR analysis targeting the σC gene. Sequencing and phylogenetic analysis showed that ARV xj-1.1 exhibited 79–81% homology with previously reported strains, suggesting that this strain represents a newly emerging variant within the circulating ARV population in China.

Pathogenicity assessment in SPF chicken embryos and chicks further confirmed the high virulence of ARV xj-1.1. Infection of SPF chicken embryos resulted in severe hemorrhagic lesions, tissue necrosis, and developmental abnormalities, ultimately leading to 100% mortality within 72–120 h post-infection. Infected embryos exhibited thickened chorioallantoic membranes and extensive bleeding, consistent with the severe pathological effects of virulent ARV strains [25,26]. Footpad inoculation of SPF chicks with ARV xj-1.1 resulted in severe viral arthritis, with visible redness, swelling, and joint inflammation. The morbidity rate reached 100%, confirming the high pathogenicity of this strain in young chickens. These findings suggest that ARV xj-1.1 is highly virulent and capable of causing severe clinical disease, which poses a significant threat to poultry production. The widespread genetic diversity and recombination potential of ARV pose a major challenge for disease control and vaccine development. While current vaccines (e.g., S1133, 1733, 2408) provide some level of protection, their efficacy is diminishing against newly emerging ARV variants [27,28]. Our study highlights the urgent need for continuous epidemiological surveillance of ARV in China to track genetic variations, pathogenic evolution, and vaccine efficacy. Future research should focus on the genetic characterization of ARV variants to understand their evolutionary trends, pathogenicity studies to determine the virulence factors associated with disease severity, development of new-generation vaccines targeting emerging ARV strains, and host–pathogen interaction studies to explore ARV immune evasion mechanisms. Given the high economic impact of ARV on poultry production, the identification and characterization of ARV xj-1.1 provide crucial data for developing effective prevention and control measures. Our findings lay the foundation for future vaccine research and molecular epidemiological studies to mitigate the ongoing threat posed by ARV infections in commercial poultry.

## 5. Conclusions

In this study, we successfully isolated and characterized ARV xj-1.1, a novel ARV strain from broilers in Xinjiang, China. Our findings demonstrate that ARV xj-1.1 exhibits strong infectivity in LMH cells, induces severe CPE, and replicates efficiently in vitro. Pathogenicity assessment in SPF chicken embryos and chicks further confirmed the high virulence of ARV xj-1.1, as evidenced by 100% embryo mortality and severe viral arthritis in chicks. Genetic analysis of ARV xj-1.1 revealed moderate homology (79–81%) with known ARV strains, indicating that this strain represents an emerging variant within the circulating ARV population. The increasing genetic diversity and recombination observed in ARV strains highlight the urgent need for continuous surveillance and vaccine development. Our study provides valuable insights into the epidemiology and molecular characteristics of ARV in China, contributing to the future design of targeted vaccines and control strategies. Further research should focus on understanding the pathogenic mechanisms of ARV, improving diagnostic tools, and developing effective vaccines to mitigate the economic impact of ARV infections on the poultry industry.

## Figures and Tables

**Figure 1 viruses-17-00499-f001:**
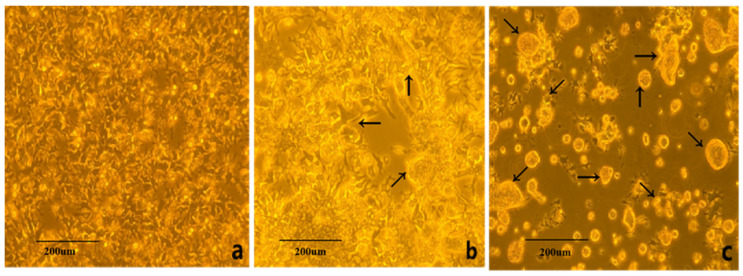
Cytopathic effects of ARV epidemic strain on LMH cells (100×). Note: (**a**). normal LMH cells; (**b**). early stage of infection (24 h); (**c**). late stage of infection (96 h). Arrow in (**b**): cells undergoing fusion and cytopathic changes. Arrow in (**c**): cells exhibiting complete fusion and cytopathic changes.

**Figure 2 viruses-17-00499-f002:**
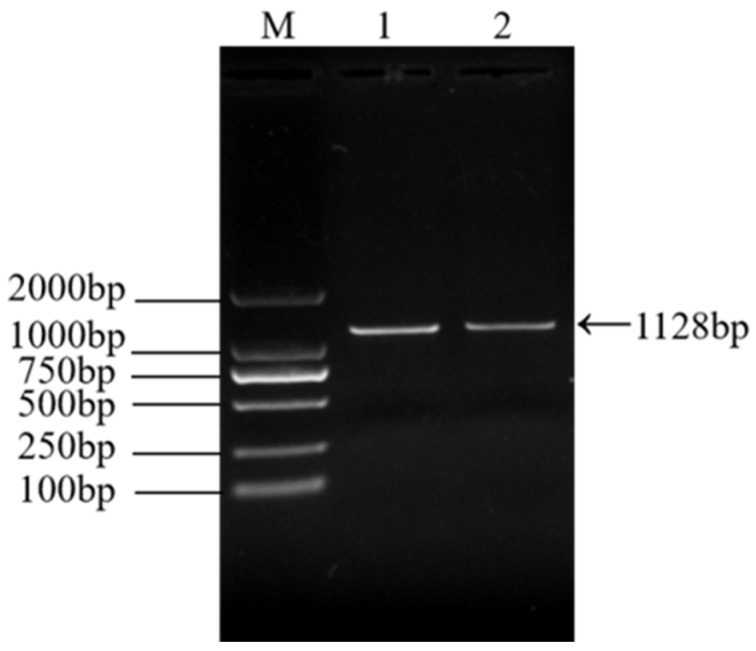
RT-PCR results of ARV epidemic strain infection in LMH cells. Note: M, Marker DL 2000; 1–2, ARV clinical isolates.

**Figure 3 viruses-17-00499-f003:**
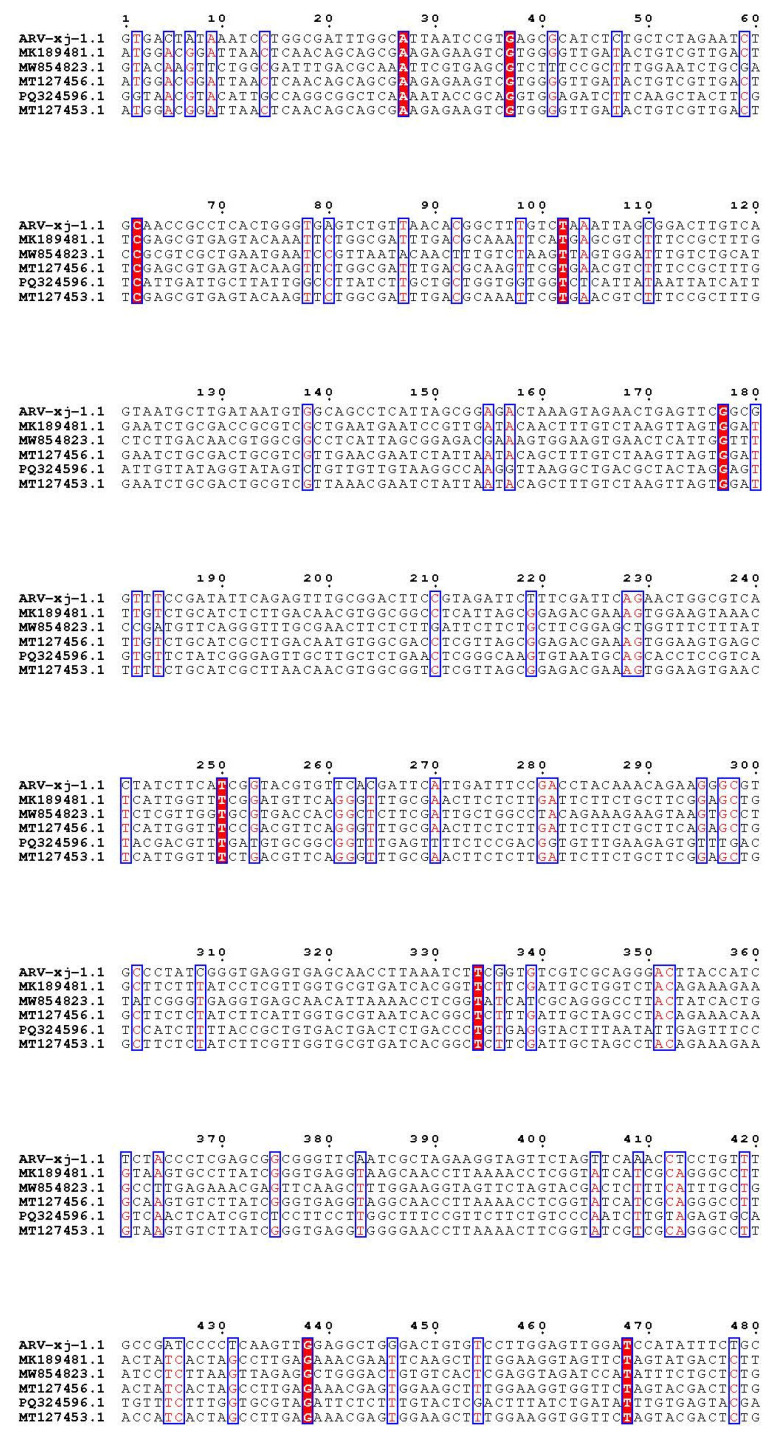
Nucleotide sequence comparison of ARV Xinjiang isolate with existing strains (partial comparison). Note: The red font text indicates the same nucleotide in one line; The box indicates that there are at least 5 nucleotides the same in one line; The line highlighted with Red background indicates the nucleotides from these 6 strains are all the same.

**Figure 4 viruses-17-00499-f004:**
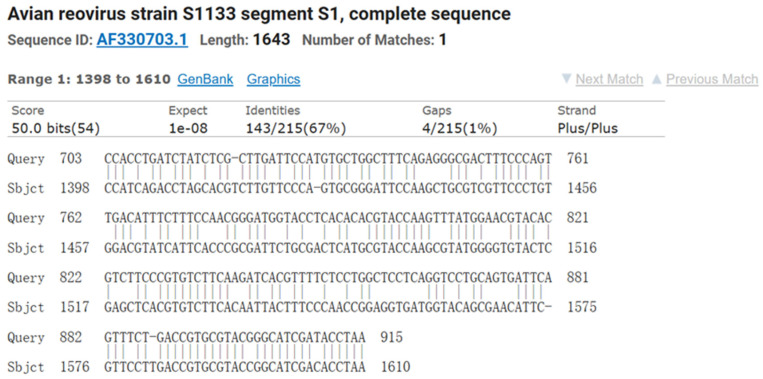
Comparison of the σC gene sequence of ARV xj-1.1 with the vaccine strain ARV S1133.

**Figure 5 viruses-17-00499-f005:**
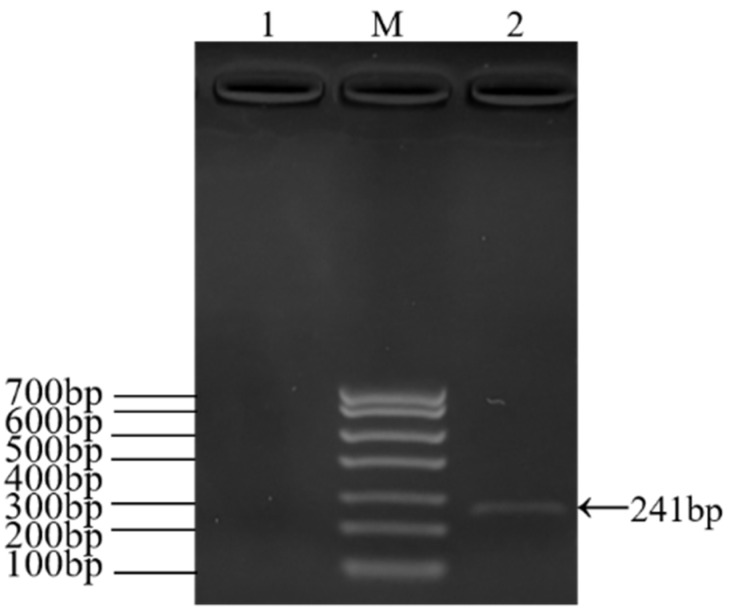
Conventional PCR detection of qPCR primers. Note: M, trans marker I; 1, ARV vaccine strain S1133; 2, ARV clinical isolate.

**Figure 6 viruses-17-00499-f006:**
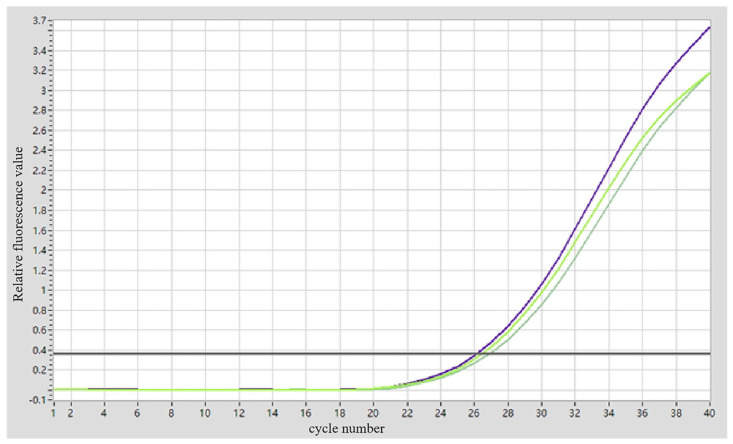
Amplification curve of the primers. Note: Different colors of the curves are triplicates of the ARV xj-1.1 samples.

**Figure 7 viruses-17-00499-f007:**
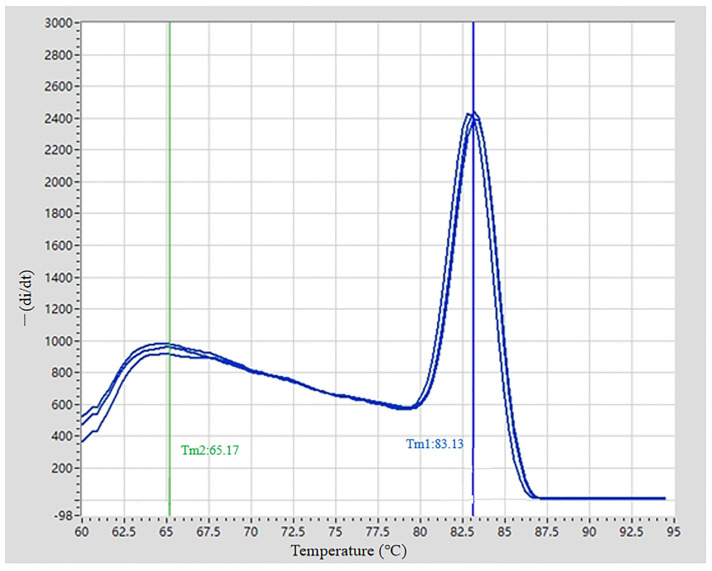
Melting curve of the qPCR primers. Note: The three blue curves are the triplicates of the ARV xj-1.1 samples; The blue line (Tm1: 83.13) is the first melting temperature—83.13 °C; The green line (Tm2: 65.17) is the second melting temperature—65.17 °C.

**Figure 8 viruses-17-00499-f008:**
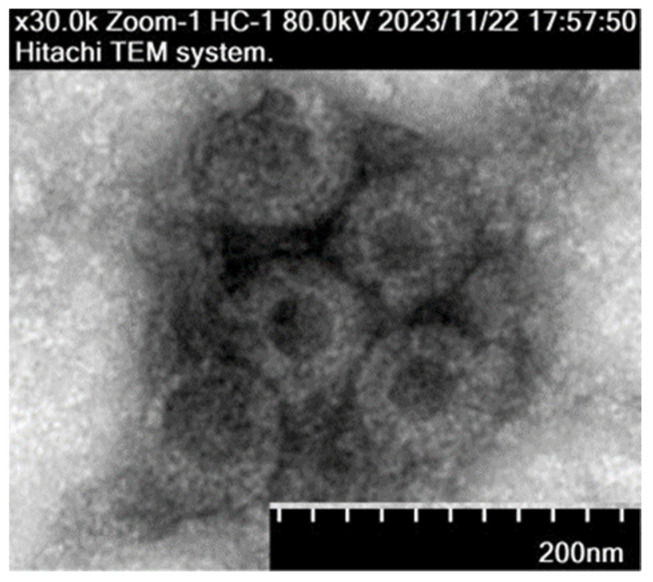
Morphology of viral particles under transmission electron microscopy (TEM).

**Figure 9 viruses-17-00499-f009:**
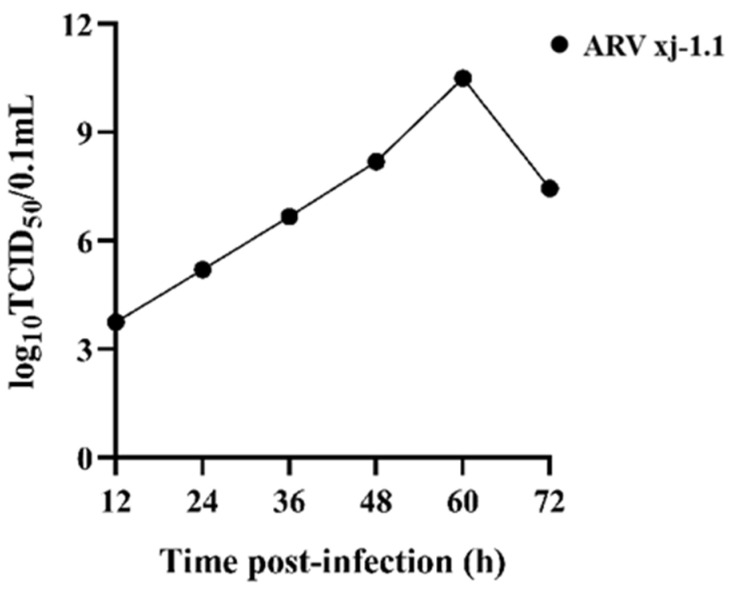
In vitro replication kinetics curve of the ARV epidemic strain.

**Figure 10 viruses-17-00499-f010:**
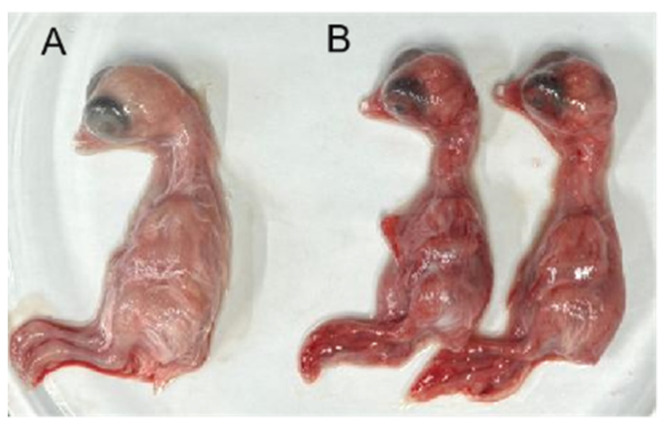
Pathogenic effects of the ARV isolate on SPF chicken embryos. Note: (**A**), uninfected chicken embryo (control group); (**B**), chicken embryo infected with ARV (pathological changes).

**Figure 11 viruses-17-00499-f011:**
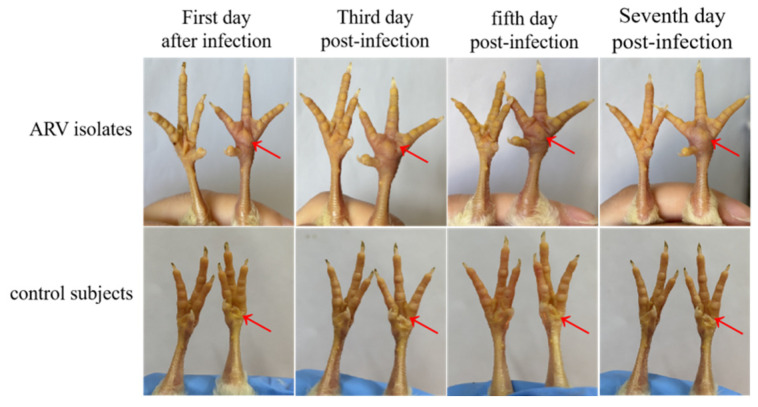
Pathogenicity test results of the ARV isolate in SPF chickens. Note: the arrows are the injection sites of different chickens. The ARV group was injected with 10^4.6^ TCID_50_/0.1 mL, and the control group was injected with 0.1 mL sterile saline.

## Data Availability

The raw data supporting the conclusions of this article will be made available by the authors upon request.

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
