# Peer review of "Isolation, Identification, and Pathogenicity of an Avian Reovirus Epidemic Strain in Xinjiang, China"

_viruses, 2025, doi:10.3390/v17040499_

Round 1

Reviewer 1 Report

Comments and Suggestions for Authors

In the manuscript entitles “Isolation, Identification, and Pathogenicity of an Avian Reovirus Epidemic Strain in Xinjiang, China” the authors have isolated and characterized chicken-derived Avian Reovirus strain (designated as ARV xj-1.1). The manuscript is generally well-addressed however, I have the following comments/ suggestions.

Line 62: since SPF chickens is used, please provide ethical statement.

Line 79: the sentence “The filtered virus solution (1 mL) was inoculated into a pre-cultured monolayer of LMH cells ......” what’s the infective dose used or just 1 ml is used?

Line 115: “F5 viral supernatant ...." please provide what is F5 refer to as this is the first time mentioned?

Line 129: provide reference for TCID50 assay.

Line 130: for growth kinetics, did you use a control to compare with?

Line 142: “Chicks were observed daily for clinical signs and footpad swelling”. provide this observation was for how long?

Line 145: Fig 1: b. Early stage of infection; c. Late stage of infection.  I suggest being specific and add the days of infection instead of early and late.

Line 161: I suggest to provide the accession numbers of the sequences of strains used in alignment with isolated strain.

Line 172: “Fluorescent Quantitative PCR Identification of ARV Virus”. This section was not mentioned in materials and methods. please revise.

Line 184: Fig 4: I suggest replacing this Figure to be as the alignment Figure (Fig. 3)

Line 331: References: please revise all references following the journal guidelines.

Author Response

Dear reviewer,

Thank you for your comments and feedback. We addressed all the comments below:

1. Line 62: since SPF chickens is used, please provide ethical statement.

Response: The ethical statement has been provided in the Institutional Review Board Statement around Line 323.

2. Line 79: the sentence “The filtered virus solution (1 mL) was inoculated into a pre-cultured monolayer of LMH cells ......” what’s the infective dose used or just 1 ml is used?

Response: The dose is 8.62*107 copies/uL, 1mL). The information has been added in the manuscript.

3. Line 115: “F5 viral supernatant ...." please provide what is F5 refer to as this is the first time mentioned?

Response: F5 refers to the fifth generation of viruses cultured on the cell culture dish. We did this to get a higher volume of viruses and get better results under electron microscopy. We have added the information to the manuscript.

4. Line 129: provide reference for TCID50 assay.

Response: The citation has been added.

5. Line 130: for growth kinetics, did you use a control to compare with?

Response: Yes, the control group was the cells without infection. We added the information to the manuscript.

6. Line 142: “Chicks were observed daily for clinical signs and footpad swelling”. provide this observation was for how long?

Response: Thank you for asking! The observation was for 7 days. We checked the chicken on days 1, 3, 5, 7. The information has been added.

7. Line 145: Fig 1: b. Early stage of infection; c. Late stage of infection.  I suggest being specific and add the days of infection instead of early and late.

Response: Thank you for the comments! Figure b is 24 hours, and c is 96 hours. We added the information to the manuscript.

8. Line 161: I suggest to provide the accession numbers of the sequences of strains used in alignment with isolated strain.

Response: Thank you for the suggestion! We have added the information to the manuscript.

9. Line 172: “Fluorescent Quantitative PCR Identification of ARV Virus”. This section was not mentioned in materials and methods. please revise.

Response: The method was provided in the Method 2.6 (Line 101).

10. Line 184: Fig 4: I suggest replacing this Figure to be as the alignment Figure (Fig. 3)

Response: Thank you for the suggestion. However, the purpose for us to make Figure 4 is different from Figure 3. We want to show that ARV Xinjiang 1.1 has a huge difference between ARV S1133.

11. Line 331: References: please revise all references following the journal guidelines.

Response: Thank you for your comment. The format has been addressed.

Reviewer 2 Report

Comments and Suggestions for Authors

Isolation, Identification, and Pathogenicity of an Avian Reovirus Epidemic Strain in Xinjian, China byt xin Ma etc is a well written manuscript. There are a few comments and suggestions that the authors should address before decision of the editor.

Line 134  How did you determine the inoculation amount? Maybe I missed the part about how you selected 104.6 TCID50 .

Line 217. How many days post inoculation did you see clinical signs? I could not tell that in the paragraph or figure 11.  Did you perform any histopathology on the 1 day old inoculated group to further document microscopic lesions that typically accompany footpad inoculation?

Line 2545-255-- The format of citation appears a little weird. Please review the format and adjust accordingly.

line 264 Change "serial blinds passages" to "serial blind passages".

References

Line 337- I believe that is the incorrect way to format a citation for a book chapter. There is no indication of volume, publisher, editor etc.

Line 348-- similar citation format issue.  Also why is an older version of Diseases of poultry being used instead of the 2020 version?

Figures:

Figure 3 is really hard to read and interpret the changes indicated by the boxes. Perhaps making that larger would help.

Figure 11. There is no explanation in Figure 11 what the arrows are pointing at. Figure legends should capture all the information without having to back into the manuscript to determine what is being shown.

Author Response

Dear reviewer:

Thank you for your comments and advice. The response is below:

1. Line 134  How did you determine the inoculation amount? Maybe I missed the part about how you selected 104.6 TCID50 .

Response: We determined the TCID50 for the chicks first, and it was 104.6, 0.1mL. That is the reason why we selected this inoculation amount.

2. Line 217. How many days post inoculation did you see clinical signs? I could not tell that in the paragraph or figure 11.  Did you perform any histopathology on the 1 day old inoculated group to further document microscopic lesions that typically accompany footpad inoculation?

Response: After one day of inoculation, we found the clinical sign. The results have been described in the Results part. However, we did not do the histopathology, because the footpad clinical sign is one of the standard methods to investigate the infection of this virus.

3. Line 2545-255-- The format of citation appears a little weird. Please review the format and adjust accordingly.

Response: Thank you for the suggestion. The format has been revised according to the journal's requirements.

4. line 264 Change "serial blinds passages" to "serial blind passages".

Response: Thank you for the suggestion. It has been revised.

5. Line 337- I believe that is the incorrect way to format a citation for a book chapter. There is no indication of volume, publisher, editor etc.

Response: Thank you for your advice. It has been revised.

6. Line 348-- similar citation format issue.  Also why is an older version of Diseases of poultry being used instead of the 2020 version?

Response: The 2020 version and this one is not written by the same author, so it is two different books.

7. Figure 3 is really hard to read and interpret the changes indicated by the boxes. Perhaps making that larger would help.

Response: The figure is the original one that we downloaded from the NCBI website. We upload another figure with the revised manuscript to the editor to see if the new one would be better.

8. Figure 11. There is no explanation in Figure 11 what the arrows are pointing at. Figure legends should capture all the information without having to back into the manuscript to determine what is being shown.

Response: Thank you for the advice. The figure legends has been added.

Reviewer 3 Report

Comments and Suggestions for Authors

This report by Ma et al. describes isolation, identification, and pathogenicity of an avian reovirus epidemic strain in Xinjiang, China. The novelty of methology and results is low since many similar reports have been published. I suggest that this report can be submitted to other more relevant journals such as Avian Disease or Avian Pathology. The quality of some photography could be improved (ie. Figures 1 &8)

Comments on the Quality of English Language

NA

Author Response

We appreciate the reviewer’s feedback regarding the alignment and figure quality. While we understand the reviewer’s perspective, we have carefully reviewed the aims and scope of the journal and confirmed that our manuscript’s objectives, particularly concerning virus isolation, identification, pathogenicity, and epidemiological relevance, align closely with the journal’s stated areas of interest. Furthermore, we have carefully reviewed all submitted figures and confirmed that they reflect the original image quality obtained during experimentation. We appreciate your valuable suggestions, and we hope our clarifications adequately address your concerns.

Round 2

Reviewer 3 Report

Comments and Suggestions for Authors

Comments as suggested for the first time. Similar reports have been published.  The novelty of findings and methods used in this study are low.   

Comments on the Quality of English Language

NA